# A Recognition Method for Marigold Picking Points Based on the Lightweight SCS-YOLO-Seg Model

**DOI:** 10.3390/s25154820

**Published:** 2025-08-05

**Authors:** Baojian Ma, Zhenghao Wu, Yun Ge, Bangbang Chen, He Zhang, Hao Xia, Dongyun Wang

**Affiliations:** 1Department of Mechanical and Electrical Engineering, Xinjiang Institute of Technology, Aksu 843100, China; 11813019@zju.edu.cn (B.M.); chenbangbang@st.xatu.edu.cn (B.C.); 2College of Mechanical and Electrical Engineering, Shihezi University, Shihezi 832003, China; wuzhenghao@stu.shzu.edu.cn (Z.W.); zhangh@stu.shzu.edu.cn (H.Z.); xiahao@stu.shzu.edu.cn (H.X.); 3College of Engineering, Zhejiang Normal University, Jinhua 321004, China

**Keywords:** marigold, segmentation, lightweight model, picking point recognition, automated harvesting

## Abstract

Accurate identification of picking points remains a critical challenge for automated marigold harvesting, primarily due to complex backgrounds and significant pose variations of the flowers. To overcome this challenge, this study proposes SCS-YOLO-Seg, a novel method based on a lightweight segmentation model. The approach enhances the baseline YOLOv8n-seg architecture by replacing its backbone with StarNet and introducing C2f-Star, a novel lightweight feature extraction module. These modifications achieve substantial model compression, significantly reducing the model size, parameter count, and computational complexity (GFLOPs). Segmentation efficiency is further optimized through a dual-path collaborative architecture (Seg-Marigold head). Following mask extraction, picking points are determined by intersecting the optimized elliptical mask fitting results with the stem skeleton. Experimental results demonstrate that SCS-YOLO-Seg effectively balances model compression with segmentation performance. Compared to YOLOv8n-seg, it maintains high accuracy while significantly reducing resource requirements, achieving a picking point identification accuracy of 93.36% with an average inference time of 28.66 ms per image. This work provides a robust and efficient solution for vision systems in automated marigold harvesting.

## 1. Introduction

Marigold (Tagetes erecta), an annual herb that bears terminal flowers, possesses significant economic value. Its extracts (e.g., lutein) are extensively utilized in food, cosmetics, pharmaceuticals, and animal feed [1]. Owing to its unique geographical location and natural conditions, both the cultivation area and yield of marigolds in Southern Xinjiang have increased annually. By 2025, the marigold cultivation area in Xinjiang had reached approximately 300,000 mu (about 20,010 hectares), primarily located in the Southern Xinjiang region. Within this area, Shache County (a county in Xinjiang) has expanded marigold cultivation to exceeding 100,000 mu (exceeding 6670 hectares) county-wide, establishing itself as the largest marigold cultivation base in China. Despite the commercial significance of marigold corollas, harvesting remains predominantly manual. This reliance on manual labor, coupled with rising labor costs, often leads to delayed operations and substantial yield losses in large-scale cultivation. Existing mechanical harvesters for terminal flowers (e.g., marigold, safflower, Chrysanthemum morifolium) primarily utilize end-effector mechanisms including cutting, roller, and comb-type systems [2]. However, non-selective harvesting of marigold corollas yields low efficiency and increases plant damage, while agricultural robotics presents a promising alternative through selective harvesting, where the precise identification of picking points is of paramount importance.

Traditional flower recognition approaches relying on color, shape, and texture features remain vulnerable to lighting variations, occlusion, adhesion, and backgrounds with similar colors, compromising recognition performance [3,4]. The increasing adoption of deep learning in agriculture has spurred researchers to adapt this technology for flower species identification. Zhang et al. [5] enhanced red flower detection accuracy by leveraging a Faster R-CNN framework with a ResNeSt-101 backbone and optimized anchor configurations, significantly improving robustness under adverse environmental conditions, including lighting variations, occlusion, and weather conditions. Subsequently, Zhang et al. [6] proposed the single-stage WED-YOLO framework, integrating WIoU loss, attention mechanisms, and dedicated small-target detection layers to enhance detection performance in complex environments. This model demonstrates superior performance over benchmark models and comparative algorithms, achieving higher accuracy, recall, and mean average precision (mAP). Additionally, to enhance floral bud and bloom detection, Zhao et al. [7] proposed an enhanced CR-YOLOv5s algorithm incorporating a coordinate attention mechanism for enriched feature representation and RepVGG blocks for optimized network architecture, significantly improving both accuracy and robustness. Although attention mechanisms are widely incorporated in enhanced models to improve accuracy, these mechanisms typically escalate model complexity and parameter requirements. In contrast, Qi et al. [8] proposed a lightweight convolutional neural network model for high-precision real-time detection of medicinal chrysanthemums in complex unstructured environments, achieving high detection accuracy for chrysanthemums and deploying the model on low-power embedded devices. Moreover, Chen et al. [9] proposed the lightweight YOLO-SaFi algorithm, incorporating structural optimizations to enable high-precision, real-time detection of red stamens in complex environments, while significantly reducing both model parameters and computational complexity.

Pruning and distillation techniques reduce model size and parameter count to achieve compression. For instance, Fan et al. [10] developed a lightweight YOLOv7 variant optimized for marigold detection. Through convolutional module refinement, network pruning, and SPP simplification, this model maintained baseline accuracy while substantially improving inference speed and deployment efficiency. Fatehi et al. [11] employed knowledge distillation to enhance YOLOv9t for real-time rose flowering detection, achieving a 0.3% increase in mean average precision (mAP@0.5), a detection speed improvement, and maintaining a compact model size of 4.43 MB. To address complex multi-stage flower detection, Zhou et al. [12] proposed a YOLOv7-based approach that achieved 94.8% mAP for real-time, in situ identification of five distinct jasmine growth stages (small bud, bud, half-open, fully open, and withered) within natural environments. Park et al. [13] developed a lightweight YOLOv4-Tiny variant utilizing circular bounding boxes, enabling efficient and accurate detection and classification of chrysanthemum flowering stages (bud, initial-bloom, and full-bloom) through architectural optimization. Zhang et al. [14] developed a drone-based method utilizing UAV-acquired RGB imagery and HSV color indices, integrating these features with a Random Forest (RF) model to classify chrysanthemum flowering stages into early, full-bloom, and late phases. Despite significant progress in multi-stage floral recognition, the accurate localization of picking points persists as a critical challenge for automated harvesting systems.

In recent years, some scholars have conducted research on the identification methods of picking points for apical flowers. Zhang et al. [15] addressed the challenge of locating safflower picking points under varying morphology and occlusion by developing the SBP-YOLOv8s-seg network, which integrates deep learning with morphological analysis techniques. This approach enhanced the mean average precision (mAP) by 1.3% over the baseline model, achieving a picking point recognition accuracy of 92.9% with a model size of 34.9 MB. Xing et al. [16] proposed a method for safflower picking point identification, based on enhanced Particle Swarm Optimization (PSO) and rotated rectangle algorithms. Their approach leveraged R-channel extraction from RGB images combined with geometric features, enabling precise stamen segmentation and localization with an accuracy of 89.75%, although the model size and parameter count were not specified. While traditional PSO approaches have been applied to safflower picking point localization, Xing et al. [17] subsequently proposed a lightweight, improved DeepLabv3+ model, achieving 92.50% localization success at 25.54 MB model size with substantially lower computational costs. Wang et al. [18] developed a two-stage recognition model incorporating an enhanced YOLOv5 architecture with Swin Transformer attention mechanisms and morphological analysis, achieving a localization accuracy of 98.19% while maintaining robust real-time performance with a lightweight structure of only 7.6 M parameters and 14.9 MB model size. Similarly, Chen et al. [19] developed a YOLOv3-based computer vision system for tea bud and picking point recognition, achieving 83% identification accuracy with a model size of 235 MB and 62 million parameters. Likewise, Li et al. [20] utilized the YOLOv3 model for efficient tea bud region detection, addressing the challenge of invisible picking points by integrating tea bud growth characteristics with a sleeve picking strategy. Their field experiments demonstrated a picking success rate of 83.18%, with a model parameter count of 61.95 M and a size of 236 MB. Owing to the distinct growth characteristics exhibited by different plant species, methodologies for identifying apical picking points vary considerably. Furthermore, current segmentation models employed for this purpose often exhibit substantial computational complexity and parameter volume, hindering their practical implementation on resource-constrained edge computing devices.

This study constructed a natural-scene marigold image dataset and introduced enhanced YOLOv8n-seg strategies to simultaneously reduce model size, parameter count, and computational complexity. The proposed methodology establishes an effective framework for precise picking point identification while preserving detection accuracy. Key innovations include:(1)To reduce the model size and parameter count, the StarNet model was employed to replace the original backbone network, and a novel lightweight feature extraction module, C2f-Star, was constructed.(2)A Seg-Marigold segmentation head featuring a dual-path collaborative architecture was designed, further minimizing the model scale while enhancing segmentation efficiency.(3)Based on the extracted corolla and stem masks, a picking point identification method was proposed, integrating corolla contour fitting with skeleton refinement techniques.

## 2. Materials and Methods

### 2.1. Image Acquisition and Data Processing

A dataset of 1847 high-resolution (4160 × 3120 pixels) marigold plant images was collected in Shache County, Xinjiang, using a Huawei P20 smartphone (Huawei Technologies Co., Ltd., Shenzhen, China). To ensure high-quality and efficient annotations for the Corolla and Stem categories, a human–machine collaborative strategy was adopted. Initially, 200 representative images were manually annotated using the open-source tool X-AnyLabeling, generating XML-format labels. These annotations were used to train a YOLOv8n model [21] to convergence. The best-performing model was converted to ONNX format and deployed for automatic annotation of the remaining images. All automatically generated labels underwent thorough manual verification and correction. The final dataset of 1847 images was partitioned into training (1293 images, 70%), validation (369 images, 20%), and test (185 images, 10%) subsets. Representative samples are shown in Figure 1b–d.

### 2.2. Overall Overview of Marigold Picking Point Recognition Method

To achieve precise localization of marigold picking points, this study introduces SCS-YOLO-Seg, a novel lightweight segmentation model (Figure 2). The model incorporates a lightweight backbone reconstruction, optimized feature extraction modules, and a redesigned segmentation head. These enhancements significantly reduce model parameters and computational complexity while maintaining high-precision segmentation performance for both corollas and stems. Following segmentation, morphological contour fitting and edge thinning algorithms are employed to accurately extract boundaries. The spatial coordinates of intersection points between the corolla and stem boundaries are then calculated based on geometric constraints. Finally, the optimal picking point is selected by applying predefined strategies that account for the specific morphological characteristics of marigolds.

### 2.3. Improvements of the YOLOv8n-Seg Model

#### 2.3.1. YOLOv8n-Seg Model

YOLOv8-seg extends the YOLOv8 object detection framework to achieve efficient image segmentation. To enhance feature extraction speed, the backbone network incorporates a lightweight design: the original C3 module is replaced by the C2f module within the CSP structure, and a Spatial Pyramid Pooling Fast (SPPF) module is integrated. Further refinements occur in the neck network’s FPN-PAN structure. Specifically, the upsampling convolutional layer is removed, and the C2f module is introduced to strengthen multi-scale feature fusion. The detection head employs a decoupled, anchor-free design, improving target localization accuracy. For segmentation tasks, YOLOv8-seg leverages concepts from YOLACT [22], dynamically generating segmentation masks through the prediction of mask coefficients. This combined strategy yields notable improvements in segmentation accuracy and processing efficiency across complex scenes, while maintaining real-time inference capabilities.

#### 2.3.2. The SCS-YOLO-Seg Segmentation Model

To achieve an optimal balance among segmentation accuracy, memory footprint, and model size for efficient edge deployment, this study proposes SCS-YOLO-Seg, an enhanced architecture based on YOLOv8n-seg (Figure 3). Key innovations focus on three critical aspects. (1) Backbone reconstruction: The original backbone is replaced with the StarNet architecture. StarNet enhances local feature extraction through spatial topology optimization within its Star block and facilitates multi-level feature fusion via a cross-scale interaction module, strengthening feature representation in complex scenes. (2) Feature fusion optimization: A lightweight C2f-Star module is introduced in the network neck, superseding the standard C2f. This module employs a feature dimension decoupling strategy and integrates depthwise separable convolution with channel recalibration. Critically, it enables dynamic screening of features across stages, substantially reducing model parameters while improving feature discriminability. (3) Segmentation head innovation: To mitigate the computational cost of the original head, a novel Seg-Marigold head is proposed. Utilizing a dual-path collaborative mechanism and a dual-branch feature decoding architecture, this head further minimizes model parameters and enhances segmentation efficiency.

#### 2.3.3. The Network Structure of StarNet

To enable lightweight deployment on edge devices while retaining marigold-detection accuracy, we replace the Darknet53 backbone of YOLOv8n with a hierarchical StarNet architecture [23]. The network comprises four sequential feature extraction stages. Each stage initiates with downsampling via a 3 × 3 convolutional layer (stride = 2), preserving channel dimensionality, followed by feature refinement using Star blocks (recursive Star Operation module within each Star block). Depthwise-separable convolutions are selectively inserted at key locations to expand receptive fields; in the final stage, their depthwise component uses a 7 × 7 kernel (stride = 1). A channel expansion factor of 4 is applied stage by stage, quadrupling the width at every transition. BatchNorm layers are replaced with LayerNorm to accelerate training. Moreover, a novel Star Operation module projects features into a high-dimensional space, fuses them via element-wise multiplication, and then re-projects—simultaneously lowering computational cost and enriching representation. As illustrated in Figure 4, this lightweight hierarchical design enables StarNet to match Darknet53 in feature-extraction quality while offering superior efficiency and a significantly smaller parameter count.

#### 2.3.4. C2f-Star Module

To optimize the parameter count and computational complexity of the C2f module within the YOLOv8 architecture, this study designs the C2f-Star module, inspired by StarNet’s lightweight design principles. Utilizing class inheritance, the module replaces the original bottleneck with a lightweight Star block. This unit employs depthwise separable convolution (DW-Conv) [24] for efficient feature extraction and constructs a transformation pathway using batch normalization (BN) layers, and dual fully connected (FC) layers, with ReLU6 activation functions applied between these layers to enhance nonlinear representation. Combined with a residual connection fusing the DW-Conv output with the original input, the Star block unit establishes an efficient feature reuse mechanism. Globally deploying the C2f-Star module within the Neck’s feature fusion stage maintains performance while significantly reducing parameters and computational load. As shown in Figure 5, multi-level lightweight strategies—spanning spatial mapping, feature reuse, and dimension transformation—enable comprehensive computational efficiency gains.

#### 2.3.5. Segmentation Head

This study proposes Seg-Marigold, a lightweight segmentation head that further reduces parameters and computational complexity (Figure 6). The module employs a hierarchical convolutional architecture during feature extraction, constructing multi-scale feature representations by cascading 1 × 1 and 3 × 3 convolutional kernels. Group Normalization (GN) [25] is integrated after each convolution, forming Conv-GN blocks that mitigate vanishing gradients and promote training stability and faster convergence. At the output stage, Seg-Marigold implements a dual-path mechanism. One path achieves pixel-level segmentation through a convolutional output channel to generate refined target contours, while the other path introduces a joint optimization strategy combining bounding box regression loss (Bbox Loss) and classification cross-entropy loss (Cls Loss) to guide segmentation accuracy. Furthermore, a scale adaptive adjustment unit (Scale) dynamically adjusts feature map dimensions according to task requirements, ensuring compatibility with multi-resolution inputs. This multi-branch collaborative architecture and multi-loss joint optimization strategy significantly improve the segmentation performance of the Seg-Marigold head.

### 2.4. Picking Point Recognition Strategy

This paper proposes a method for identifying the picking points of marigolds based on contour fitting and skeleton thinning (Figure 7), which consists of four core steps:(1)Image segmentation and mask binarization: We processed a segmented marigold image to precisely extract masks corresponding to the corolla (category index 0) and stem (category index 1), obtaining their distinct regions of interest (ROIs). These masks were then binarized to isolate clear target regions for subsequent morphological analysis, specifically corolla ellipse fitting and stem characterization.(2)Corolla ellipse fitting and optimization (circles in the Figure 7): To accurately define the marigold corolla picking region and simulate its natural morphology, we fit an ellipse to the binary corolla mask. Reflecting the natural tilted orientation observed in the corolla, the initial fitting used an inclination angle of 30°. Additionally, to incorporate a safety margin during harvesting—preventing corolla damage and reducing stem detachment risk—we scaled the minor axis of the fitted ellipse by a factor of 1.4. This optimization expanded the ellipse’s coverage of the target corolla region, improving the reliability of picking-point identification.(3)Skeleton extraction of the stem: To extract the stem skeleton, the contour was first refined using morphological opening and closing operations to eliminate small noise points and fill minor holes, thereby enhancing structural integrity and continuity. Subsequently, the Zhang–Suen thinning algorithm [26] was applied to the processed binary region to obtain a preliminary central skeleton. As the initial skeleton often contains extraneous small branches that are not representative of the primary stem structure and could potentially complicate downstream analysis, connected component analysis was employed to identify and prune branches shorter than 10 pixels. This step ensured the retention of only the core stem skeleton.(4)Determination of picking point (dots in the Figure 7): The intersection between the optimized corolla ellipse and the stem skeleton was determined by performing a pixel-wise comparison of the binary stem skeleton image and the ellipse mask image. An intersection was registered at pixel coordinates where both images exhibited non-zero values. If the initial 30° oriented ellipse yielded no intersections, the ellipse was rotated by 90°, and the comparison was repeated. If intersections were found after rotation, the point with the minimum Y-coordinate (the uppermost position in the image coordinate system) was selected as the optimal picking point. The procedure terminated if no intersections were detected after the 90° rotation.

**Figure 7 sensors-25-04820-f007:**
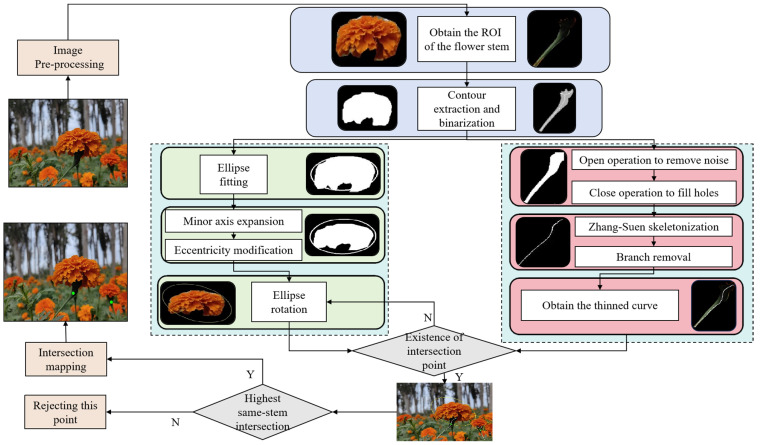
Picking Point Recognition Strategy.

### 2.5. Experimental Environment and Model Evaluation Indicators

#### 2.5.1. Experimental Environment Configuration

To ensure experimental reproducibility and environmental consistency, all calculations were performed on a uniformly configured computing platform. The hardware comprised an Intel Core i9 processor and a GPU with 16 GB of memory. The operating system was Windows 10. The software environment was established using Anaconda 3.8 (Anaconda Inc., Austin, TX, USA), with Python 3.9. Model implementation utilized PyTorch 2.2, with acceleration provided by CUDA 12.1.

#### 2.5.2. Evaluation Metrics of Model

To comprehensively quantify the performance and efficiency of the SCS-YOLO-Seg model, this study employs six key evaluation metrics: precision (P), recall (R), mean average precision (mAP), model size, parameter count, and floating-point operations (GFLOPs). Precision (P) denotes the proportion of correctly identified positive instances among all instances predicted as positive, while recall (R) represents the proportion of actual positive instances successfully detected. The mean average precision (mAP), integrating precision across varying recall thresholds, serves as the primary metric for assessing overall segmentation accuracy. Model size directly reflects structural complexity and storage requirements. Parameter count and computational complexity, measured in GFLOPs, collectively determine the model’s computational efficiency and memory footprint, constituting critical parameters for evaluating deployment feasibility and inference speed.

In the context of marigold harvesting point identification, the precise localization of picking points is defined as the detection of the calyx and its adjacent regions. A picking point is considered accurately detected only when the model correctly identifies this target area; otherwise, the localization is deemed unsuccessful. To quantitatively evaluate model recognition performance, this study employs accuracy (*AP*, Accuracy Precision) and misjudgment rate (*WP*, Wrong-point Rate) as evaluation metrics.(1)$$AP=\frac{Point\_a}{Point\_all}\times 100\%$$(2)$$WP=\frac{Point\_e}{Point\_all}\times 100\%$$

Point_all denotes the total number of target points identified for picking. Within this set, Point_a indicates the number of points accurately detected, while Point_e represents the number of points that were target points but remained undetected or unrecognized during the picking process.

## 3. Results

### 3.1. Model Training

During training of the proposed SCS-YOLO-Seg model, we employed the default hyperparameters of the original YOLOv8n architecture: an input resolution of 640 × 640 pixels, a batch size of 32, an initial learning rate of 0.01, and Stochastic Gradient Descent (SGD) as the optimizer. Training proceeded for 300 epochs. Figure 8a illustrates the evolution of the training loss and mean average precision (mAP). The loss function exhibits a consistent decreasing trend over the iterations, accompanied by a corresponding increase in mAP. Analysis of Figure 8b reveals that both the training loss and validation mAP metrics stabilized, indicating convergence, around epoch 200. The smooth decline in loss and steady rise in mAP throughout training, culminating in stable convergence, demonstrates an effective optimization process. Furthermore, the absence of significant divergence between the training and validation metrics suggests that overfitting did not occur.

### 3.2. Comparison Experiment of Lightweight Models

This study uses YOLO v8n as the benchmark model. By comparing StarNet with mainstream lightweight models such as MobileNetV4 [27], FasterNet [28], EfficientViT [29], and RepViT [30], this study demonstrates StarNet’s superior ability to balance segmentation accuracy with model efficiency. Specifically, in the corolla category segmentation task, StarNet attains a precision of 79.2% and an mAP@0.5 of 89.1%, surpassing comparable models by 0.2% and 0.9% in mAP@0.5 over EfficientViT and MobileNetV4, respectively. For stem category segmentation, StarNet achieves a precision of 81.6%, a recall of 68.8%, and an mAP@0.5 of 78.6%, exhibiting the strongest overall segmentation performance. Although RepViT yields a marginally higher recall (71.1%), its large model size (13.8 MB) and computational cost (22.4 GFLOPs) significantly compromise its lightweight utility. In contrast, StarNet maintains the most compact model footprint, minimal parameterization, and lowest computational overhead among all evaluated architectures (Table 1). Overall, StarNet achieves an optimal balance between accuracy and efficiency, offering enhanced practicality and deployment advantages for resource-constrained scenarios.

### 3.3. Ablation Experiment

We conducted ablation studies on the YOLOv8n baseline model (Table 2) to evaluate the effectiveness of our proposed module enhancements for corolla and stem segmentation. Replacing the original backbone for our StartNet brought significant gains in corolla segmentation: precision (P) jumped to 79.2% and mAP@0.5 reached 89.1%. Concurrently, the model slimmed down considerably: model size was reduced by 20.0%, parameters by 24.3%, and computational cost (GFLOPs) by 13.3%. However, we observed a slight trade-off with a minor decline in stem segmentation performance. Building upon the StartNet backbone, we incorporated the C2f-Star module, which yielded further reductions in size, parameters, and GFLOPs, though accompanied by slight decreases in segmentation accuracy for both classes.

Similarly, the integration of the Seg-Marigold module with the StartNet backbone substantially compressed the model to 3.5 MB (a 46.2% reduction) and reduced GFLOPs to 8.4, while marginally increasing corolla precision to 78.4%. Corolla recall (R) and mAP@0.5 remained comparable to baseline levels, and stem accuracy exhibited a small drop. The final enhanced model, integrating all three modules (StartNet, C2f-Star, Seg-Marigold), achieved an optimal balance between segmentation performance and model efficiency. For the corolla class, precision (P) increased from 77.1% to 78.7%, while recall (R) decreased marginally by 0.8% and mAP@0.5 declined only slightly by 0.5%. Stem class metrics (P, R, mAP@0.5) decreased by 1.2%, 1.9%, and 1.7%, respectively. Crucially, this configuration achieved substantial model compression: model size was reduced by 52.3%, parameters by 53.7%, and GFLOPs by 32.5%. These results demonstrate that the proposed method effectively balances significant model compression with segmentation performance, maintaining high accuracy with only minimal compromises.

### 3.4. Comparison of Segmentation Performance for Different Models

To evaluate the performance of the SCS-YOLO-Seg model in the marigold segmentation task, we compared it with the YOLOv5n [31], YOLOv8n, YOLOv9c [32], YOLOv11n [33], and YOLOv12n [34] models on the same dataset. As summarized in Table 3, SCS-YOLO-Seg achieved a precision (P) of 78.7%, a recall (R) of 82.6%, and mAP@0.5 of 88.0% for the corolla category. While its precision was 1.4% lower than YOLOv5n (80.1%), recall was 3.3% lower than YOLOv9c (85.9%), and mAP@0.5 was 0.9% lower than YOLOv9c (88.9%), its primary advantage lies in its extremely lightweight design. For the Stem category, SCS-YOLO-Seg attained P, R, and mAP@0.5 values of 82.3%, 67.4%, and 78.0%, respectively: 1.2% lower in precision than YOLOv8n (83.5%), 4.9% lower in recall than YOLOv9c (72.3%), and 1.3% lower in mAP@0.5 than YOLOv9c (79.3%). Critically, SCS-YOLO-Seg requires only 3.1 MB in model size (versus 56.3 MB for YOLOv9c) and 8.1 GFLOPs computational cost (versus 157.6 GFLOPs for YOLOv9c). Thus, despite marginally lower segmentation metrics in certain categories compared to top performers, SCS-YOLO-Seg maintains high accuracy while drastically reducing computational complexity and parameter scale, demonstrating strong potential for resource-constrained deployment scenarios.

Figure 9 illustrates segmentation performance on marigold images under varying lighting conditions (front light, back light, side light). Under front illumination, all models achieve satisfactory corolla segmentation, but both YOLOv8 and YOLOv12 fail to consistently detect stems (red arrows). Similar stem segmentation failures occur under backlighting. Sidelighting poses a greater challenge: segmentation accuracy for the corolla region drops considerably in YOLOv5, YOLOv8, and YOLOv12 models (red arrows). In contrast, the proposed SCS-YOLO-Seg model maintains effective segmentation of both corolla and stem components across all lighting conditions. Critically, its segmentation quality shows minimal sensitivity to the lighting direction (front, back, or side), indicating strong robustness to illumination variations.

To visually evaluate the segmentation performance of the proposed SCS-YOLO-Seg model on the marigold dataset, we employed Grad-CAM [35], a technique for generating visual explanations. The resulting heatmaps highlight the model’s key regions of interest. As shown in Figure 10, the model achieved high segmentation accuracy for both the corolla and stem across three representative lighting conditions. The heatmaps exhibit strong activation responses (highlighted areas) corresponding to these structures, in clear contrast to the low-activation background (blue regions). This demonstrates that the model effectively focuses on and segments the target organs from the background. These findings confirm the capability of SCS-YOLO-Seg to extract discriminative features for marigolds, leading to accurate localization of the spatial positions and morphological boundaries of the corolla and stem. This precise segmentation establishes a critical foundation for subsequent picking point recognition tasks.

### 3.5. Picking Point Recognition Result

The performance of the proposed marigold picking point recognition method was evaluated on a test set comprising 185 images (Table 4), containing 572 manually annotated ground truth points. The method detected 534 points correctly, resulting in 38 missed detections and achieving a recognition accuracy of 93.36% (equivalent to an error rate of 6.64%). The average processing time per image was 28.66 ms, with preprocessing accounting for 2.53 ms and model inference requiring 26.73 ms. Figure 11 presents representative results under diverse lighting conditions, depicting key stages of the process: mask segmentation, flower crown fitting, stem skeletonization, and the final localization of picking points. The method demonstrates robust performance in generating accurate flower crown segmentation masks and localizing picking points consistently across varying illumination. Despite the high accuracy in flower crown segmentation, the segmentation of slender stems presents a significant challenge. Crucially, occlusion of the target stem by the flower crown or adjacent leaves can compromise stem segmentation. This impairment frequently leads to failures in stem extraction, which is the primary cause of the observed missed detections for picking points.

## 4. Discussion

Current research on automated harvesting predominantly focuses on recognizing picking points for terminal flowers like safflower [15,18], typically employing segmentation-based methods on image or point cloud data. However, improving segmentation accuracy often relies on computationally intensive approaches, such as attention mechanisms or deeper architectures, which substantially increase model complexity and parameter count, hindering deployment on resource-constrained edge devices. To overcome this limitation, we proposed SCS-YOLO-Seg, a lightweight and efficient segmentation model for identifying marigold picking points. The model achieved accurate corolla and stem segmentation with significantly reduced computational demands, featuring only 1.5 million parameters, 3.1 MB storage occupancy, and 8.1 GFLOPs. Leveraging this efficient segmentation, we further developed a novel picking point recognition method by integrating contour fitting and skeleton thinning, achieving a recognition accuracy of 93.36%. This approach demonstrates applicability to automating the harvesting of similar terminal flowers, such as edible roses.

The segmentation of marigold stems within complex field environments presents a significant challenge for accurate picking point localization. This difficulty arises from the collective interference of weed and leaf occlusion, the inherent morphological slenderness of the stems, and their chromatic similarity to the background, which substantially hinders robust contour extraction in Figure 12a,b. Furthermore, adherence or occlusion of the stem by the corolla frequently leads to erroneous or missed identification of picking points, even when corolla segmentation is successful (Figure 12c). In Figure 12d, occlusion from an adjacent plant’s stem prevented complete segmentation of a target marigold plant within a complex background, leading to inaccurate picking point identification. Similarly, Figure 12e demonstrates that occlusion by the corolla of another marigold plant adversely affected stem segmentation, resulting in the unrecognized picking point of this specimen. Notably, viewpoint variations induced by robotic movement [36] can reveal previously occluded stems, enabling effective picking point identification by the proposed method under these conditions. In future work, we will further investigate cross-task feature interaction mechanisms to address erroneous segmentation arising from various occlusion patterns [37].

The dataset used in this study primarily consists of samples captured under normal lighting conditions (frontlight, backlight, and sidelight). However, it lacks representation of extreme weather scenarios, such as rain, fog, or high winds. Under these unrepresented conditions, elliptical fitting accuracy for corollas (e.g., estimating tilt angles) is likely to degrade, and stem skeleton extraction may be less complete. This degradation could consequently affect the stability of the picking point identification. Additionally, data collection was confined to the marigold cultivation area in Shache County, Xinjiang. As a result, the dataset reflects only the growth morphology of this specific local cultivar and does not include samples from other varieties. Therefore, the model’s generalization capability to unseen marigold cultivars may be limited.

Subsequent research will assess marigold maturity based on corolla size or color to enable selective harvesting. This strategy retains immature flowers, allowing for future harvests. Depth cameras will be used to acquire marigold image datasets. By integrating depth information with flower recognition algorithms, the system can precisely locate corolla picking points. Setting an effective distance threshold for the depth camera automatically filters out distant plant data, effectively mitigating picking point misidentification caused by distant outliers and reducing computational load, thereby enhancing efficiency. Additionally, leveraging cross-view information fusion (e.g., combining RGB and depth imagery) can further improve the robustness of marigold picking-point detection under complex field conditions [38].

## 5. Conclusions

This study presents SCS-YOLO-Seg, a lightweight segmentation architecture engineered for robust recognition of marigold picking points within automated agricultural systems. Building upon the YOLOv8n framework, the model integrates three core enhancements: (1) the Starnet backbone, optimized for feature extraction efficiency; (2) the novel C2f-Star module, designed to enhance feature fusion capabilities; and (3) a specialized Seg-Marigold head, tailored specifically for the precise segmentation of picking points.

The primary contribution of this work is the achievement of an optimal balance between model efficiency and segmentation accuracy. Comprehensive evaluations demonstrate significant model compression: compared to the YOLOv8n baseline, SCS-YOLO-Seg reduces model size by 52.3%, parameters by 53.7%, and computational load (FLOPs) by 32.5%. Importantly, these efficiency gains are attained without compromising segmentation performance. The model achieves a corolla segmentation mAP@0.5 of 88.0%, closely matching the performance of the considerably larger YOLOv9c-seg model (88.9%). For stem segmentation, SCS-YOLO-Seg achieves an mAP@0.5 of 78.0%, comparable to or exceeding other efficient YOLO variants. Furthermore, the proposed method enables high-precision localization of marigold picking points (AP: 93.36%, error rate: 6.64%) through morphological analysis of segmentation masks. Its computational efficiency, with a mean processing time of 28.66 ms per image, confirms its robustness and suitability for real-time field deployment on resource-constrained platforms.

Despite persistent challenges such as occlusion, SCS-YOLO-Seg demonstrates significant potential as an efficient and accurate solution for automated marigold harvesting. Its compact model size and low computational complexity are particularly advantageous for practical implementation on agricultural robotic platforms, while maintaining a high level of segmentation accuracy crucial for reliable operation. Future research will investigate methods to enhance robustness under heavy occlusion, integrate depth sensing for precise 3D localization, and leverage visual cues for maturity detection to support selective harvesting strategies.

## Figures and Tables

**Figure 1 sensors-25-04820-f001:**
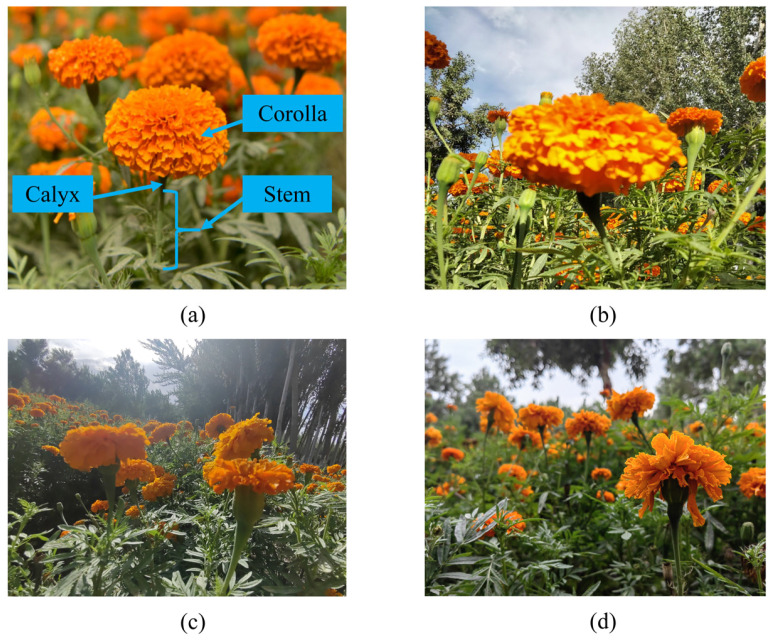
Data of marigold samples. (**a**) The morphology of marigold plants; (**b**) front lighting; (**c**) back lighting; (**d**) side lighting.

**Figure 2 sensors-25-04820-f002:**
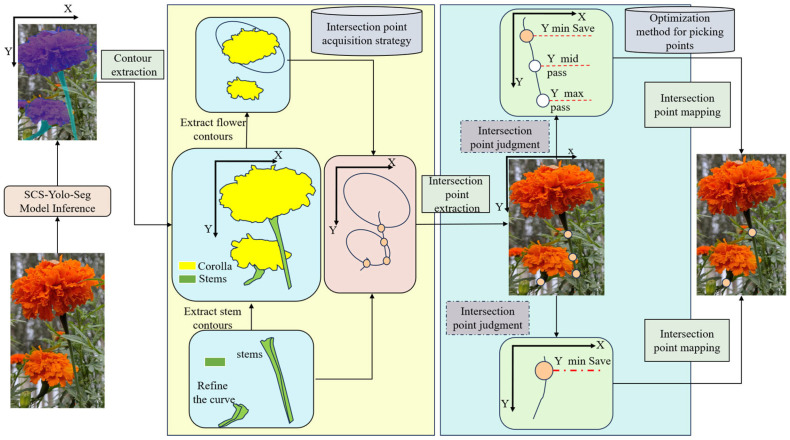
Overall process for identifying marigold picking points.

**Figure 3 sensors-25-04820-f003:**
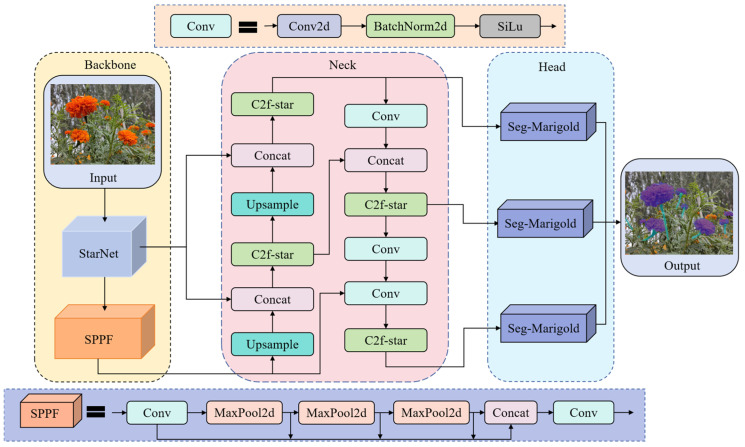
Network structure of SCS-YOLO-Seg.

**Figure 4 sensors-25-04820-f004:**
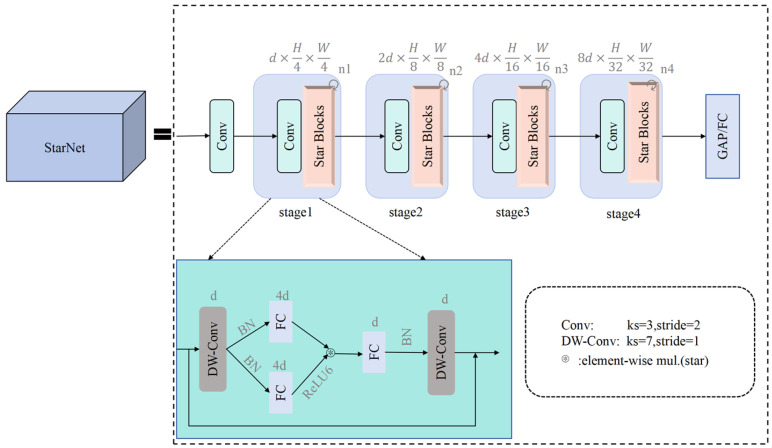
StarNet structure.

**Figure 5 sensors-25-04820-f005:**
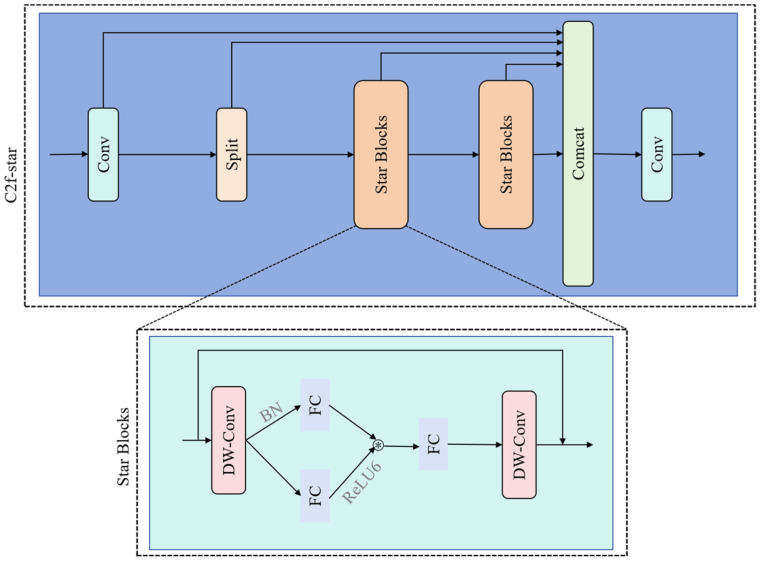
C2f-star structure diagram.

**Figure 6 sensors-25-04820-f006:**
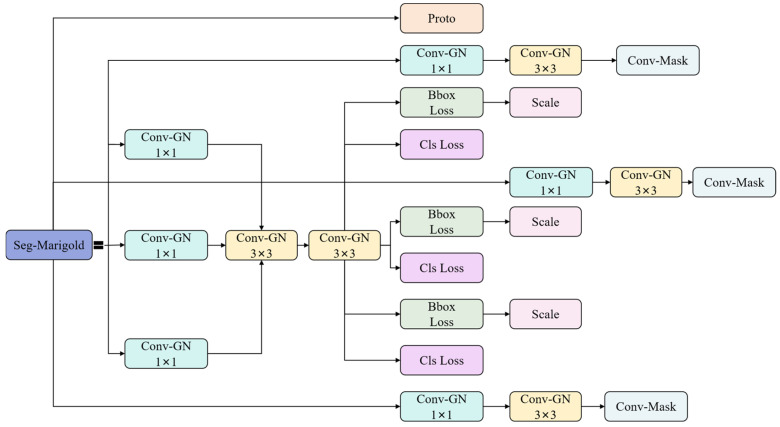
Segmentation head.

**Figure 8 sensors-25-04820-f008:**
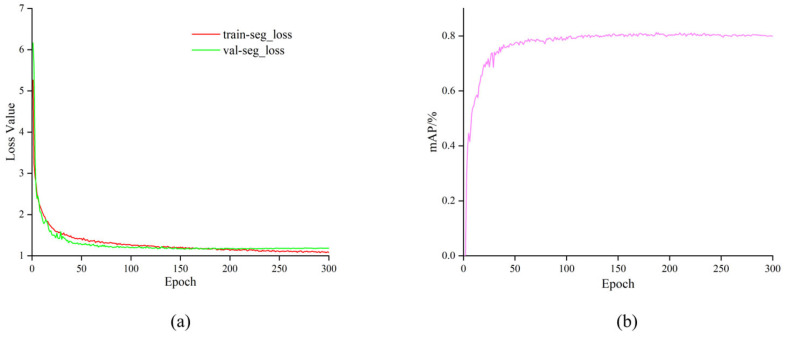
Training results. (**a**) loss curve; (**b**) mAP.

**Figure 9 sensors-25-04820-f009:**
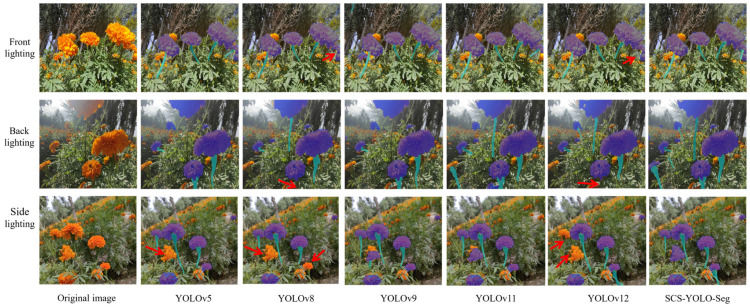
The segmentation results of different models.

**Figure 10 sensors-25-04820-f010:**
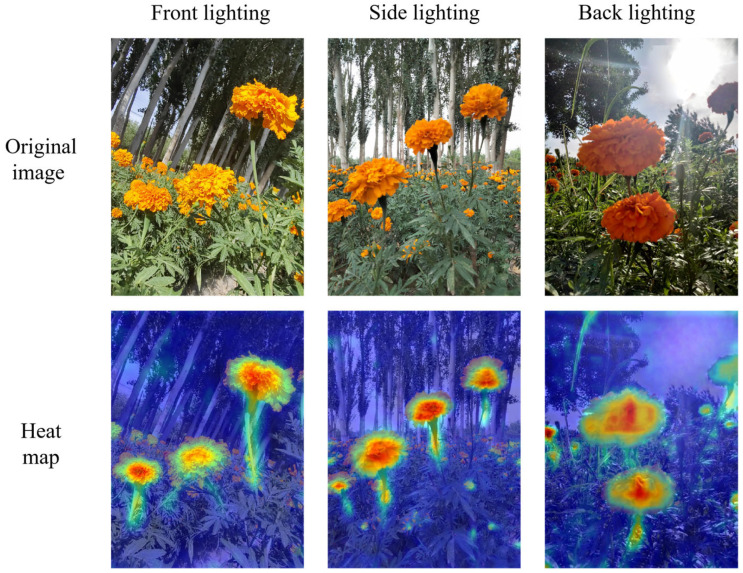
Heatmap of marigold segmentation based on the SCS-YOLO-Seg model.

**Figure 11 sensors-25-04820-f011:**
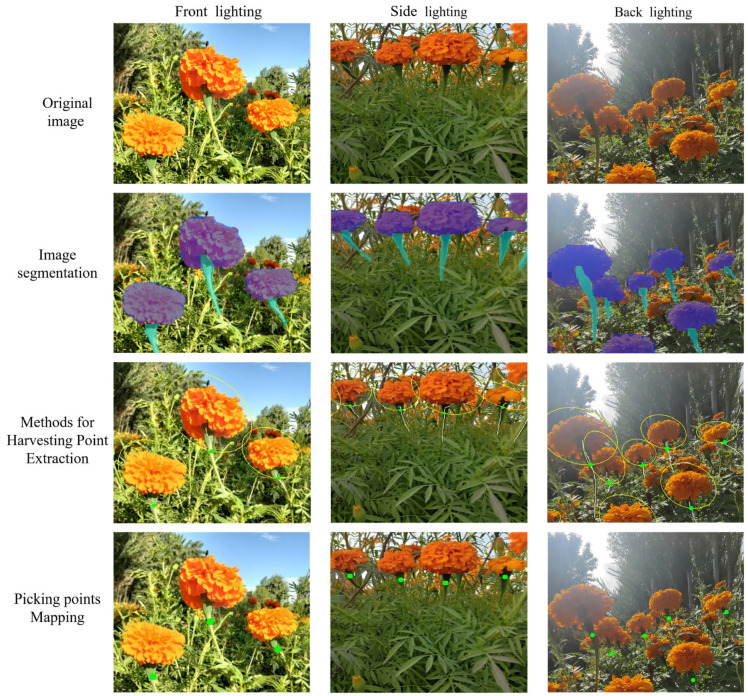
The recognition process of picking points with 3 types of lighting.

**Figure 12 sensors-25-04820-f012:**
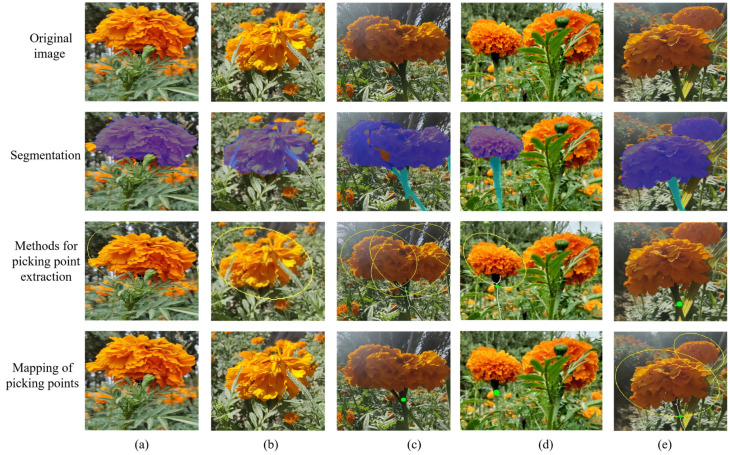
Picking point recognition error. (**a**,**b**) Occlusion of weed and leaf; (**c**,**d**) occlusion of the stem; (**e**) occlusion of corolla.

**Table 1 sensors-25-04820-t001:** Performance evaluation of lightweight backbone architectures.

Backbone	Category	P (%)	R (%)	mAP@0.5(%)	Model Size (MB)	Parameter Count	GFLOPs
EfficientViT	corolla	78.0	84.6	88.9	8.8	4,261,078	13.30
	stem	77.8	68.1	77.5			
MobileNetV4	corolla	76.1	84.0	88.2	11.7	5,952,694	26.4
	stem	78.9	67.9	77.4			
Fastnet	corolla	79.2	83.5	89.1	8.7	4,424,986	14.6
	stem	81.3	68.8	78.6			
Repvit	corolla	77.8	82.4	87.8	13.8	4,470,816	22.4
	stem	81.1	71.1	78.6			
Starnet	corolla	79.2	83.5	89.1	5.2	2,465,590	10.4
	stem	81.6	68.8	78.6			

**Table 2 sensors-25-04820-t002:** Ablation results.

Baseline Model	Starnet	C2f-Star	Seg-Marigold	Category	P (%)	R (%)	mAP@0.5 (%)	Model Size (MB)	Parameter Count	GFLOPs
YOLOv8n	×	×	×	corolla	77.1	83.4	88.5	6.5	3,258,454	12
				stem	83.5	69.3	79.7			
YOLOv8n	√	×	×	corolla	79.2	83.5	89.1	5.2	2,465,590	10.4
				stem	81.6	68.8	78.6			
YOLOv8n	√	√	×	corolla	76.6	82.5	88.0	4.6	2,265,430	10
				stem	80.3	69.4	76.9			
YOLOv8n	√	×	√	corolla	78.4	82.1	87.9	3.5	1,707,509	8.4
				stem	77.7	67.4	76.1			
YOLOv8n	√	√	√	corolla	78.7	82.6	88	3.1	1,507,349	8.1
				stem	82.3	67.4	78			

**Table 3 sensors-25-04820-t003:** The segmentation performance of different models.

Different Models	Category	P (%)	R (%)	mAP@0.5(%)	Model Size (MB)	Parameter Count	GFLOPs
YOLOv5n	corolla	80.1	82.2	87.8	3.9	1,881,103	6.7
	stem	81.0	69.3	77.0			
YOLOv9c	corolla	74.7	85.9	88.9	56.3	27,626,070	157.6
	stem	79.6	72.3	79.3			
YOLOv11n	corolla	76.5	85.4	88.8	5.8	2,834,958	10.2
	stem	79.3	72.4	79.4			
YOLOv12n	corolla	74.7	83.1	87.6	5.7	2,761,150	9.2
	stem	81.8	67.8	77.4			
YOLOv8n	corolla	77.1	83.4	88.5	6.5	3,258,454	12.0
	stem	83.5	69.3	79.7			
SCS-YOLO-Seg	corolla	78.7	82.6	88.0	3.1	1,507,349	8.1
	stem	82.3	67.4	78.0			

**Table 4 sensors-25-04820-t004:** Picking point recognition result.

Segmentation Model	Point_all	Point_a	Point_e	AP (%)	WP (%)	Composition of Time (ms)		
						Preprocessing	Reasoning	Identification
SCS-YOLO-Seg	572	534	38	93.36	6.64	2.53	26.73	28.66

## Data Availability

The original contributions presented in this study are included in the article; further inquiries can be directed to the corresponding author.

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
