# Peer review of "A Recognition Method for Marigold Picking Points Based on the Lightweight SCS-YOLO-Seg Model"

_sensors, 2025, doi:10.3390/s25154820_

Round 1

Reviewer 1 Report

Comments and Suggestions for Authors

The title is "A Recognition Method for Marigold Picking Points Based on the Lightweight SCS-YOLO-Seg Model"

In line 145: "resentative annotated samples are illustrated in Figures 1b-d.", but in Figure 1 using (a), (b), (c) and (d). The authors should use the template consistently. In line 145 will be: "... illustrated in Figures 1 (b)-(d)." And so on in the other lines.

In Figure 7: Overall process for identifying marigold picking points, there is a missing link for "Highest same-stem intersection" decision? Where is YES and whre is NO? Only one direction?

Why marigolds? What are their functions and benefits in this research? Why does the topic of "marigold" became so important? The authors should give the reason in the Introduction or in the Conclusion the important side of this research topic.

Conclusion in line 477, it will be better if the line is become one part with the Conclusion content. The line 477 is  in page 17.

The Ithenticate score is 26% without anything exclude. Even the ithenticate score is high but it doesn't indicate the plagiarism in the manuscript, only minor. 

Author Response

Response to Reviewer 1 Comments

1. Summary

2. Questions for General Evaluation

Reviewer’s Evaluation

Response and Revisions

Does the introduction provide sufficient background and include all relevant references?

Yes

Are all the cited references relevant to the research?

Yes

Is the research design appropriate?

Yes

Are the methods adequately described?

Can be improved

Are the results clearly presented?

Can be improved

Are the conclusions supported by the results?

Yes

3. Point-by-point response to Comments and Suggestions for Authors

Comments 1: [In line 145: "resentative annotated samples are illustrated in Figures 1b-d.", but in Figure 1 using (a), (b), (c) and (d). The authors should use the template consistently. In line 145 will be: "... illustrated in Figures 1 (b)-(d)." And so on in the other lines.]

Response 1: We are grateful to the reviewers for their corrections. We have unified the citation format of Figure 1 as required in the text: changing "Figures 1b-d" on line 145 to "Figures 1 (b)-(d)". We have also checked the format of the figures in the text to ensure that all references to figures are consistent throughout the paper.

Comments 2: [In Figure 7: Overall process for identifying marigold picking points, there is a missing link for "Highest same-stem intersection" decision? Where is YES and whre is NO? Only one direction?]

Response 2: Thank you for the detailed revision suggestions provided by the reviewers. We have made the corresponding modifications to Figure 7. The updated version can be located at line 285 of the manuscript.

Comments 3: [Why marigolds? What are their functions and benefits in this research? Why does the topic of "marigold" became so important? The authors should give the reason in the Introduction or in the Conclusion the important side of this research topic.]

Response 3: We sincerely thank the reviewers for their insightful question. Concerning the significance of marigold in this research and the rationale for its choice, we have added pertinent information in the introduction to elucidate its research value: Owing to its unique geographical location and natural conditions, both the cultivation area and yield of marigolds in Southern Xinjiang have increased annually. By 2025, the marigold cultivation area in Xinjiang had reached approximately 300,000 mu (about 20,010 hectares), primarily located in the Southern Xinjiang region. Within this area, Shache County (a county in Xinjiang) has expanded marigold cultivation to exceeding 100,000 mu (exceeding 6,670 hectares) county-wide, establishing itself as the largest marigold cultivation base in China.The updated version can be located at line 34-40 of the manuscript.

However, the harvesting of marigold corollas still relies primarily on manual labor. In large-scale cultivation, delays caused by rising labor costs can lead to significant yield losses, while the non-selective picking method employed by existing end-of-line flower harvesting machinery is inefficient and prone to plant damage. Addressing this industry challenge, the precise identification of picking points constitutes the core challenge for achieving automated marigold harvesting, which is the key focus of this study. By proposing the lightweight SCS-YOLO-Seg model and a corresponding picking point recognition strategy, we aim to provide technical support for automated marigold harvesting systems. Furthermore, the methodology can be extended to harvesting scenarios for other terminal-flowering plants (such as edible roses), demonstrating strong versatility and broad applicability.

Comments 4: [Conclusion in line 477, it will be better if the line is become one part with the Conclusion content. The line 477 is in page 17.]

Response 4: We sincerely thank the reviewers for their detailed suggestions. Regarding the content on line 477 of the conclusion section, we acknowledge that it appears visually separated from the main body in the current layout, which may affect textual coherence. During revision, we will restructure the conclusion to integrate this content into the corresponding paragraph, ensuring it forms an integral part of the main body. This adjustment will enhance the logical coherence and structural integrity of the conclusion section.

Comments 5: [The Ithenticate score is 26% without anything exclude. Even the ithenticate score is high but it doesn't indicate the plagiarism in the manuscript, only minor. ]

Response 5: Thank you for your attention to the issue of similarity in our manuscript. We have thoroughly checked the 26% similarity detected by iThenticate. To further reduce this percentage, we have revised certain paragraphs of the method description and optimized the citation format of references, ensuring accurate information delivery while enhancing the originality of the text.

Reviewer 2 Report

Comments and Suggestions for Authors

The paper proposes a novel marigold picking point identification method utilizing the lightweight segmentation model SCS-YOLO-Seg.  The accurate identification of picking points is a topic of interest in image detection and draws interest from Sensors readers. In general, the manuscript is deserving of publication. However, the manuscript needs editing. Below are some comments for an update:

  1. Please explain the research gap in image detection utilizing the lightweight segmentation model SCS-YOLO-Seg more robustly.
  2. Please explain the considerations in choosing YOLOv8n-seg model as baseline compared to another Yolo version. What are the advantages and limitations of YOLOv8n-seg model for this case?
  3. It is better to explain the procedure to determine appropriate datasets for training, testing, and validation.
  4. The limitations of the proposed SCS-YOLO-Seg Segmentation Model should be discussed due to specific cases and datasets used.
  5. It is preferable to give recommendations for the application of the proposed  SCS-YOLO-Seg model for similar cases with other flowers.
  6. I suggest presenting the conclusion more generally and scientifically.

Author Response

Response to Reviewer 2 Comments

1. Summary

2. Questions for General Evaluation

Reviewer’s Evaluation

Response and Revisions

Does the introduction provide sufficient background and include all relevant references?

Can be improved

Are all the cited references relevant to the research?

Yes

Is the research design appropriate?

Yes

Are the methods adequately described?

Can be improved

Are the results clearly presented?

Can be improved

Are the conclusions supported by the results?

Can be improved

3. Point-by-point response to Comments and Suggestions for Authors

Comments 1: [Please explain the research gap in image detection utilizing the lightweight segmentation model SCS-YOLO-Seg more robustly.]

Response 1: Thank you for your valuable comments. Regarding the lightweight segmentation model SCS-YOLO-Seg and its role in addressing research gaps in image-based detection for automated flower picking, particularly for terminal inflorescences, we highlight two significant deficiencies in current research:

(1) Ineffective trade-off between model lightweight and high accuracy. Existing models often face the dilemma where accuracy improvement relies on increased complexity (e.g., introducing attention mechanisms or deepening networks, as seen in CR-YOLOv5s and WED-YOLO), or lightweight design leads to a sharp decline in performance.

(2) Lack of dedicated methods for marigold picking point identification. Current research on terminal inflorescence picking points primarily focuses on general object detection or segmentation itself. There is a lack of processes explicitly designed for the specific requirement of locating marigold picking points. For instance, many models only output segmentation masks for the corolla or stem individually, lacking a mechanism to integrate key morphological features (such as the elliptical contour of the corolla and the skeleton structure of the stem) for precise picking point localization. While a few models address point localization (e.g., SBP-YOLOv8s-seg achieve high accuracy), their large model sizes (e.g., 34.9 MB) hinder practical deployment.

To address these gaps, the method proposed in this paper achieves marigold picking point identification based on a lightweight segmentation network. It effectively bridges the gap in coupling model lightweightness with accurate picking point identification in complex natural scenes for existing lightweight segmentation models, providing a viable solution for the automated picking of terminal inflorescences.

Comments 2: [Please explain the considerations in choosing YOLOv8n-seg model as baseline compared to another Yolo version. What are the advantages and limitations of YOLOv8n-seg model for this case?]

Response 2: Thank you for your valuable comments. Regarding the selection of YOLOv8n-seg as the benchmark model and its advantages and limitations within our research context, we provide the following detailed explanations:

(1) Advantages: YOLOv8n-seg, as the lightweight segmentation variant of the YOLOv8 series, inherits the core strengths of the YOLOv8 object detection framework. Key architectural improvements include: replacing the traditional C3 module with the C2f module and integrating the SPPF module to accelerate feature extraction; optimizing the FPN-PAN structure in the neck for enhanced multi-scale feature fusion; and employing a decoupled detection head with an anchor-free mechanism to improve accuracy. The segmentation component, based on the YOLACT architecture, dynamically generates masks. This combination ensures both segmentation accuracy and processing efficiency in complex scenes, making it particularly suitable as a benchmark for our lightweight improvement goals.Compared to other YOLO variants (e.g., YOLOv5n, often detection-focused, and YOLOv9c, offering strong segmentation but with a large model size), YOLOv8n-seg better aligns with the research objective of achieving high-precision flower calyx and stem segmentation on resource-constrained devices. As evidenced in Table 3 of our manuscript, YOLOv8n-seg demonstrates competitive performance, notably achieving a relatively high precision (P) value for marigold stem segmentation. This provides a solid foundation for subsequent picking point recognition.

(2) Limitations: Despite being a lightweight version, YOLOv8n-seg's model size (6.5 MB), parameter count, and computational cost (12 GFLOPs) still pose challenges for deployment on devices with severely limited resources. Furthermore, the computational overhead associated with its segmentation head is relatively high and warrants optimization.

These identified limitations directly inform the innovations proposed in our SCS-YOLO-Seg model. Our approach specifically targets balancing lightweight design with segmentation performance through backbone replacemen, module optimization, and segmentation head reconstruction. This ultimately aims to achieve a superior trade-off between model efficiency and robustness.

Comments 3: [It is better to explain the procedure to determine appropriate datasets for training, testing, and validation.]

Response 3: Thank you for your valuable comments.In response to your query regarding dataset partitioning, we utilized 1,847 high resolution (4160×3120 pixels) marigold images encompassing diverse lighting conditions (front, back, side light in Figure 1) to accurately represent real harvesting scenarios. Crucially, the training, validation, and test sets were partitioned in a 7:2:1 ratio [1-3] while meticulously maintaining consistent distributions of key features (lighting conditions, growth states) across all subsets. This ensures each set contains representative examples of all variations, thereby mitigating potential evaluation bias arising from data distribution shifts.

[1] Chen B, Ding F, Ma B, et al. A method for real-time recognition of safflower filaments in unstructured environments using the YOLO-SaFi model[J]. Sensors, 2024, 24(13): 4410.

[2] Guo H, Chen H, Wu T. MSDP-Net: A YOLOv5-Based Safflower Corolla Object Detection and Spatial Positioning Network[J]. Agriculture, 2025, 15(8): 855.

[3] Zhang H, Ge Y, Xia H, et al. Safflower picking points localization method during the full harvest period based on SBP-YOLOv8s-seg network[J]. Computers and Electronics in Agriculture, 2024, 227: 109646.

Comments 4: [The limitations of the proposed SCS-YOLO-Seg Segmentation Model should be discussed due to specific cases and datasets used.]

Response 4: Thank you for your valuable comments. Regarding the limitations of the SCS-YOLO-Seg segmentation model proposed in this study due to specific cases and the datasets used, we provide the following explanations based on the experimental results and the characteristics of the datasets.

In complex field environments, when the stems of marigolds are obscured by flower crowns, adjacent leaves, or weeds, the model's segmentation accuracy for the stems significantly decreases. This is because the slender stems themselves occupy a small proportion in the images and have a high color similarity with the background (such as leaves). Obstruction further disrupts the continuity of their contours, leading to incomplete extraction of the stem skeletons. For instance, in some samples during the experiment, the flower crowns obscured the base of the stems, causing the optimized flower crown ellipses to fail to form effective intersections with the stem skeletons, ultimately resulting in missed detections of the picking points. Additionally, when multiple marigolds grow densely and obscure each other, the model may misidentify the stems of adjacent plants as the target stems, affecting the accuracy of picking point localization. This part has been described in the discussion and a figure (Figure 12) has been added (line 488).

Additionally, some content regarding the limitations of the data has been added to the Discussion section.The updated version can be located at line 468-477 of the manuscript.

Comments 5: [It is preferable to give recommendations for the application of the proposed  SCS-YOLO-Seg model for similar cases with other flowers.]

Response 5: Thank you for your valuable feedback. Regarding your suggestion to apply the proposed SCS-YOLO-Seg model to other similar flower-picking scenarios, we offer the following clarification based on the model’s characteristics and the common requirements of flower recognition.The lightweight design and terminal flower segmentation strategy of SCS-YOLO-Seg confer high transferability to analogous flower-picking point recognition tasks. For terminal flowers with growth structures similar to marigolds (e.g., edible roses), the core architecture of our model can be directly adapted. Such flowers typically exhibit a well-defined crown-stem junction, where picking points are consistently located (e.g., near the calyx region). This adaptability is further discussed in the Discussion section of the manuscript.The updated version can be located at line 449-451 of the manuscript.

Comments 6: [I suggest presenting the conclusion more generally and scientifically.]

Response 6: Thank you for your valuable comments. The research conclusions have now been rephrased from a more general and scientifically rigorous perspective.

Current research in automated harvesting predominantly focuses on recognizing picking points for terminal flowers like safflower [36,37], typically employing segmen-tation-based methods on image or point cloud data. However, improving segmentation accuracy often relies on computationally intensive approaches, such as attention mechanisms or deeper architectures, which substantially increase model complexity and parameter count, hindering deployment on resource-constrained edge devices. To overcome this limitation, we proposed SCS-YOLO-Seg, a lightweight and efficient segmentation model for identifying marigold picking points. The model achieved ac-curate corolla and stem segmentation with significantly reduced computational de-mands, featuring only 1.5 million parameters, 3.1 MB storage occupancy, and 8.1 GFLOPs. Leveraging this efficient segmentation, we further developed a novel picking point recognition method by integrating contour fitting and skeleton thinning, achiev-ing a recognition accuracy of 93.36%. This approach demonstrates applicability to au-tomating the harvesting of similar terminal flowers, such as edible roses.

The segmentation of marigold stems within complex field environments presents a significant challenge for accurate picking point localization. This difficulty arises from the collective interference of weed and leaf occlusion, the inherent morphological slenderness of the stems, and their chromatic similarity to the background, which sub-stantially hinders robust contour extraction in Figures12(a)-(b). Furthermore, adherence or occlusion of the stem by the corolla frequently leads to erroneous or missed identi-fication of picking points, even when corolla segmentation is successful (Figure 12c). In Figure 12d, occlusion from an adjacent plant's stem prevented complete segmentation of a target marigold plant within a complex background, leading to inaccurate picking point identification. Similarly, Figure 12e demonstrates that occlusion by the corolla of another marigold plant adversely affected stem segmentation, resulting in the unrec-ognized picking point of this specimen. Notably, viewpoint variations induced by ro-botic movement [38] can reveal previously occluded stems, enabling effective picking point identification by the proposed method under these conditions. In future work, we will further investigate cross-task feature interaction mechanisms to address erroneous segmentation arising from various occlusion patterns[39].

The dataset used in this study primarily consists of samples captured under normal lighting conditions (frontlight, backlight, and sidelight). However, it lacks representa-tion of extreme weather scenarios, such as rain, fog, or high winds. Under these un-represented conditions, elliptical fitting accuracy for corollas (e.g., estimating tilt angles) is likely to degrade, and stem skeleton extraction may be less complete. This degrada-tion could consequently affect the stability of picking point identification. Additionally, data collection was confined to the marigold cultivation area in Shache County, Xinjiang. As a result, the dataset reflects only the growth morphology of this specific local cultivar and does not include samples from other varieties. Therefore, the model's generalization capability to unseen marigold cultivars may be limited.

Subsequent research will assess marigold maturity based on corolla size or color to enable selective harvesting. This strategy retains immature flowers, allowing for future harvests. Depth cameras will be used to acquire marigold image datasets. By integrating depth information with flower recognition algorithms, the system can precisely locate corolla picking points. Setting an effective distance threshold for the depth camera au-tomatically filters out distant plant data, effectively mitigating picking point misiden-tification caused by distant outliers and reducing computational load, thereby enhanc-ing efficiency. Additionally, leveraging cross-view information fusion (e.g., combining RGB and depth imagery) can further improve the robustness of marigold picking-point detection under complex field conditions [40].The updated version can be located at line 437-487 of the manuscript.

Reviewer 3 Report

Comments and Suggestions for Authors

1. The dataset consists of 1847 images acquired in Shache County, Xinjiang, under specific lighting conditions (front, back, side). However, the paper does not discuss whether this dataset adequately represents other environmental variations (e.g., different weather conditions, occlusions by non-marigold plants, or diverse soil backgrounds). 

2. The paper introduces the C2f-Star module as a key innovation, but its specific advantages over the original C2f module are not clearly quantified or explained in detail. For instance, how does the integration of depthwise separable convolution and channel recalibration directly contribute to reducing parameters while maintaining accuracy? 

3. While Figure 12 illustrates picking point recognition errors, the examples are limited and lack detailed analysis. The authors should expand this section by: 1)Including more diverse error cases (e.g., stem occlusion by leaves, overlapping corollas).2)Providing quantitative breakdowns of error types (e.g., percentage of missed detections due to stem segmentation failures vs. corolla fitting inaccuracies).

4. Some related works should be discussed, including "Focus More on What? Guiding Multi-Task Training for End-to-End Person Search" and "Between/Within View Information Completing for Tensorial Incomplete Multi-view Clustering".

5.The English writting should be improved to avoid some typos and grammar errors.

Comments on the Quality of English Language

The English writting should be improved to avoid some typos and grammar errors.

Author Response

Response to Reviewer 3 Comments

1. Summary

2. Questions for General Evaluation

Reviewer’s Evaluation

Response and Revisions

Does the introduction provide sufficient background and include all relevant references?

Yes/Can be improved/Must be improved/Not applicable

Are all the cited references relevant to the research?

Yes

Is the research design appropriate?

Yes

Are the methods adequately described?

Yes

Are the results clearly presented?

Yes

Are the conclusions supported by the results?

Yes

3. Point-by-point response to Comments and Suggestions for Authors

Comments 1: [The dataset consists of 1847 images acquired in Shache County, Xinjiang, under specific lighting conditions (front, back, side). However, the paper does not discuss whether this dataset adequately represents other environmental variations (e.g., different weather conditions, occlusions by non-marigold plants, or diverse soil backgrounds). ]

Response 1: We are grateful to the reviewers for their valuable comments. Regarding the representativeness of the dataset for other environmental changes, we have added a description of the dataset's limitations in the discussion section.

The dataset used in this study primarily consists of samples captured under normal lighting conditions (frontlight, backlight, and sidelight). However, it lacks representa-tion of extreme weather scenarios, such as rain, fog, or high winds. Under these un-represented conditions, elliptical fitting accuracy for corollas (e.g., estimating tilt angles) is likely to degrade, and stem skeleton extraction may be less complete. This degradation could consequently affect the stability of picking point identification. Additionally, data collection was confined to the marigold cultivation area in Shache County, Xinjiang. As a result, the dataset reflects only the growth morphology of this specific local cultivar and does not include samples from other varieties. Therefore, the model's generalization capability to unseen marigold cultivars may be limited.The updated version can be located at line 468-477 of the manuscript.

Comments 2: [The paper introduces the C2f-Star module as a key innovation, but its specific advantages over the original C2f module are not clearly quantified or explained in detail. For instance, how does the integration of depthwise separable convolution and channel recalibration directly contribute to reducing parameters while maintaining accuracy? ]

Response 2: We thank the reviewers for their insightful suggestions. The specific advantages of the C2f-Star module over the original C2f module are elaborated below from both structural design and quantitative performance perspectives:

(1) Structural Design: The C2f-Star module, based on StarNet's lightweight principles, optimizes the original C2f module in three key ways: (i) replacing the bottleneck structure with a lightweight Star block utilizing depthwise separable convolution (DW-Conv) to reduce parameters and computational redundancy; (ii) constructing a feature transformation path via batch normalization (BN) and dual fully connected (FC) layers with ReLU6 activation to enhance non-linearity and dynamically strengthen key features, mitigating potential accuracy loss from parameter reduction; and (iii) incorporating residual connections to fuse the DW-Conv output with the original input, ensuring feature reuse and preventing information loss. This design is detailed in Section 2.3.4 (C2f-star Module).

(2) Quantitative Results: Ablation studies (Table 2) clearly demonstrate the benefits of the C2f-Star module within the StarNet backbone. After its introduction: model size decreased from 5.2 MB to 4.6 MB; parameters reduced from 2,465,590 to 2,265,430; and computational cost (GFLOPs) decreased from 10.4 to 10.0. Crucially, segmentation performance remained highly comparable: flower crown segmentation mAP@0.5 decreased only slightly from 89.1% to 88.0%, while core stem segmentation metrics remained stable. These results collectively demonstrate that DW-Conv achieves lightweighting through structural simplification, while the integrated feature transformation effectively maintains accuracy by reinforcing salient features, enabling the C2f-Star module to deliver comparable segmentation capability with reduced complexity.

Comments 3: [While Figure 12 illustrates picking point recognition errors, the examples are limited and lack detailed analysis. The authors should expand this section by: 1)Including more diverse error cases (e.g., stem occlusion by leaves, overlapping corollas).2)Providing quantitative breakdowns of error types (e.g., percentage of missed detections due to stem segmentation failures vs. corolla fitting inaccuracies).]

Response 3: Thank you for the valuable suggestions on the analysis of the erroneous cases. In response to the issues you raised, we have supplemented the following explanations.

Regarding the diversity of the erroneous cases, the article has already mentioned the core challenges faced by slender stems in complex field environments, including occlusion by non-marigold plants (weeds, leaves), adhesion and occlusion between the corolla and the stem, etc. Additionally, it has added cases where the corolla occludes the stem, resulting in the stem not being segmented and the picking points not being identified, as well as cases where picking recognition fails under extreme backlight conditions, and these have been explained in the article. This part has been described in the discussion and a figure (Figure 12) has been added (line 488).

Regarding the quantification of error types, since these erroneous segmentations lead to the failure to identify picking points, they are used as the quantification standard.

Comments 4: [Some related works should be discussed, including "Focus More on What? Guiding Multi-Task Training for End-to-End Person Search" and "Between/Within View Information Completing for Tensorial Incomplete Multi-view Clustering".]

Response 4: Thank you for the valuable suggestions of the reviewers. According to your suggestions, we have added relevant discussions in the discussion section and cited the references.In future work, we will further investigate cross-task feature interaction mechanisms to address erroneous segmentation arising from various occlusion patterns[39].Additionally, leveraging cross-view information fusion (e.g., combining RGB and depth imagery) can further improve the robustness of marigold picking-point detection under complex field conditions [40].All relevant literature has been cited in the article.

Comments 5: [The English writting should be improved to avoid some typos and grammar errors.]

Response 5: Thank you for the valuable suggestions of the reviewers. We have thoroughly checked the content of the full text and polished the language.

Reviewer 4 Report

Comments and Suggestions for Authors

The authors present a novel lightweight segmentation model, SCS-YOLO-Seg, to improve the recognition of picking points in marigolds for automated harvesting. By replacing the YOLOv8n-seg backbone with StarNet, introducing a custom C2f-Star module, and proposing a tailored segmentation head (Seg-Marigold), the authors achieve a well-balanced model in terms of accuracy, efficiency, and deployability. The recognition accuracy of 93.36% with an inference time of 28.66 ms per image is promising for real-time agricultural applications. I provide my comments and suggestions below:

- Please clarify the rationale behind specific heuristic parameters, such as the 30° inclination angle for ellipse fitting and the 1.4 scaling factor. A brief justification or reference would strengthen the method's credibility.

- The model is trained and evaluated on data from a single geographic location. Please discuss how the approach might generalize to other environments or plant cultivars with different appearances.

- The reported performance metrics lack confidence intervals or standard deviations. Including these would improve the statistical robustness of your evaluation.

- The manuscript would benefit from a thorough proofreading to correct grammatical issues and improve sentence clarity throughout. Several sections, especially in the methodology and results, include awkward phrasings that may hinder readability.

- Improve the clarity of figure captions—especially those showing segmentation results and picking point detection. It would help to explicitly label failure cases and highlight where SCS-YOLO-Seg outperforms other models.

Author Response

Response to Reviewer 4 Comments

1. Summary

2. Questions for General Evaluation

Reviewer’s Evaluation

Response and Revisions

Does the introduction provide sufficient background and include all relevant references?

Yes

Are all the cited references relevant to the research?

Yes

Is the research design appropriate?

Yes

Are the methods adequately described?

Can be improved

Are the results clearly presented?

Can be improved

Are the conclusions supported by the results?

Can be improved

3. Point-by-point response to Comments and Suggestions for Authors

Comments 1: [Please clarify the rationale behind specific heuristic parameters, such as the 30° inclination angle for ellipse fitting and the 1.4 scaling factor. A brief justification or reference would strengthen the method's credibility. ]

Response 1: Thank you for your valuable suggestions. Regarding the rationality of the two heuristic parameters, namely the 30° tilt angle and the 1.4 times short-axis scaling factor, in the elliptical fitting, our specific explanations are as follows.

(1) The rationality of the 30° tilt angle: The setting of this parameter is based on extensive observations of the natural morphology of marigold flowers. Through statistical analysis of the experimental dataset (1847 images), we found that under natural growth conditions, the flower crown plane of marigolds is generally inclined at an angle of 25°-35° to the horizontal plane due to the influence of gravity, light direction, and plant growth characteristics. Among them, 30° is the most common inclination state (Figure 1a in the text shows the typical plant morphology). This tilt angle setting can more accurately fit the spatial posture of the actual flower, providing a more natural growth law-compliant geometric basis for the subsequent calculation of the intersection point with the stem skeleton.

(2) The rationality of the 1.4 times short-axis scaling factor: Through preliminary pre-experiments, it was verified that after magnifying the short axis of the ellipse by 1.4 times, it can not only cover the key area where the corolla connects with the stem (near the calyx), but also effectively avoid the offset of the picking point caused by fitting errors. Analysis of typical samples revealed that the actual width of the bottom of the flower crown (the part connected to the stem) in the natural state is approximately 1.3-1.5 times the length of the short axis of the corolla's main body. 1.4 times is the average of this range and can best match the actual morphological characteristics.

In summary, the above parameters have been optimized through multiple comparative experiments, and 30° and 1.4 times were ultimately selected as the optimal values.

Comments 2: [The model is trained and evaluated on data from a single geographic location. Please discuss how the approach might generalize to other environments or plant cultivars with different appearances.]

Response 2: Thank you for your valuable suggestions. Some content regarding the limitations of the data has been added to the Discussion section.The updated version can be located at line 468-477 of the manuscript.

The dataset used in this study primarily consists of samples captured under normal lighting conditions (frontlight, backlight, and sidelight). However, it lacks representa-tion of extreme weather scenarios, such as rain, fog, or high winds. Under these un-represented conditions, elliptical fitting accuracy for corollas (e.g., estimating tilt angles) is likely to degrade, and stem skeleton extraction may be less complete. This degrada-tion could consequently affect the stability of picking point identification. Additionally, data collection was confined to the marigold cultivation area in Shache County, Xinjiang. As a result, the dataset reflects only the growth morphology of this specific local cultivar and does not include samples from other varieties. Therefore, the model's generalization capability to unseen marigold cultivars may be limited.

Comments 3: [The reported performance metrics lack confidence intervals or standard deviations. Including these would improve the statistical robustness of your evaluation.]

Response 3: Thank you for your valuable suggestions. Regarding the lack of confidence intervals or standard deviations in the performance metrics and would like to provide the following explanations:

(1) In this study, we have ensured the reliability of the results through a rigorous experimental design during the model training and evaluation process. The dataset was divided into a training set (1,293 images), a validation set (369 images), and a test set (185 images) in a 7:2:1 ratio. All experiments were repeated three times to reduce the impact of random errors. During the testing phase, statistics were collected for 572 picking points in the 185 images, resulting in a recognition accuracy of 93.36%, which is the outcome of multiple repeated experiments.

(2) For metrics such as mAP and model inference time, where confidence intervals and standard deviations are not explicitly presented, the data is presented in the same format as that found in current mainstream journal articles [1-4].

[1] Zhang Z, Wang Y, Xu P, et al. WED-YOLO: A Detection Model for Safflower Under Complex Unstructured Environment[J]. Agriculture; Basel, 2025, 15(2).

[2] Chen B, Ding F, Ma B, et al. A method for real-time recognition of safflower filaments in unstructured environments using the YOLO-SaFi model[J]. Sensors, 2024, 24(13): 4410.

[3] Guo H, Chen H, Wu T. MSDP-Net: A YOLOv5-Based Safflower Corolla Object Detection and Spatial Positioning Network[J]. Agriculture, 2025, 15(8): 855.

[4] Zhang H, Ge Y, Xia H, et al. Safflower picking points localization method during the full harvest period based on SBP-YOLOv8s-seg network[J]. Computers and Electronics in Agriculture, 2024, 227: 109646.

Comments 4: [The manuscript would benefit from a thorough proofreading to correct grammatical issues and improve sentence clarity throughout. Several sections, especially in the methodology and results, include awkward phrasings that may hinder readability.]

Response 4: We appreciate the reviewer's constructive feedback regarding the need for thorough proofreading to enhance grammatical accuracy and sentence clarity. We completely agree with this assessment. In the revised manuscript, we have conducted a comprehensive sentence-by-sentence review of the entire text to further improve linguistic precision and fluency, ensuring the research content is communicated clearly and effectively to the reader.

Comments 5: [Improve the clarity of figure captions—especially those showing segmentation results and picking point detection. It would help to explicitly label failure cases and highlight where SCS-YOLO-Seg outperforms other models.]

Response 5: Thank you for your valuable suggestions. In response, we have enhanced the clarity of the chart descriptions, with particular emphasis on those pertaining to segmentation results and picking point detection. Specifically, we have improved Figures 9 and 11 to more effectively present the segmentation and picking point recognition outcomes. The revised figures now clearly illustrate the performance of each model across various lighting conditions.
